# Chemical Content of Five Molluscan Bivalve Species Collected from South Korea: Multivariate Study and Safety Evaluation

**DOI:** 10.3390/foods10112690

**Published:** 2021-11-03

**Authors:** Jelena Mutić, Vesna Jovanović, Liesbeth Jacxsens, Jannes Tondeleir, Petar Ristivojević, Sladjana Djurdjić, Andreja Rajković, Tanja Ćirković Veličković

**Affiliations:** 1Faculty of Chemistry, University of Belgrade, Studentski trg 12-16, 11000 Belgrade, Serbia; jmutic@chem.bg.ac.rs (J.M.); vjovanovic@chem.bg.ac.rs (V.J.); ristivojevic@chem.bg.ac.rs (P.R.); sladjanadj@chem.bg.ac.rs (S.D.); 2Department of Molecular Biotechnology, Environmental Technology and Food Technology, Ghent University Global Campus, Incheon 21985, Korea; 3Department of Food Technology, Safety and Health, Faculty of BioScience Engineering, Ghent University, B-9000 Ghent, Belgium; Liesbeth.Jacxsens@UGent.be (L.J.); jannes.tondeleir@ugent.be (J.T.); Andreja.Rajkovic@UGent.be (A.R.); 4Department of Chemical and Biological Sciences, Serbian Academy of Sciences and Arts, 11000 Belgrade, Serbia

**Keywords:** scallop, clam, multivariate analysis, metal exposure assessment, ICP MS

## Abstract

Bivalves are a good source of nutrients but also a potential source of environmental contaminants, which could pose a risk for consumers. The aims of this study were: the determination of 16 elements by ICP-MS in 48 samples of five bivalve species purchased from market in Korea; the identification of elements useful for species classification using multivariate analyses; and the benefit-risk evaluation associated to the consumption of these bivalves. The highest difference among content of elements between species was found for Cd, Mn, Ni, Zn, and Fe. Partial last squares discriminant analysis revealed elements with a VIP score >1 which were considered as the most relevant for explaining certain species. As, Cd, Co, and Ni were found as taxonomical markers of *V. philippinarum*; Mn, Zn, Mg, and Na of *A. irradians*; and Cd, Ni, and Fe of *M. yessoensis*. These species could serve as good dietary sources of essential elements. Cd exposure by consumption of Manila clams is not representing a health risk for the Korean population; however, through consumption of Yesso scallops, 5.3% of the Korean population has a potential health risk. Removal of the digestive gland before eating will drastically reduce the amount of Cd ingested.

## 1. Introduction

Bivalves have high nutritional value, and they are considered a good source of proteins, lipids, and minerals [1]. Beside nutrients, bivalves could be a potential source of environmental organic and inorganic contaminants because they belong to the filter feeder group of animals, which have a tendency to accumulate pollutants in their body. Therefore, bivalves are considered to be good bioindicators for both metal and organic contaminants [2,3,4,5,6]. Therefore, their consumption could pose a risk for consumers’ health. This risk has increased in the last several decades due to the increasing global demand for bivalves, which relies on farm production of bivalves. In highly developed industrial countries, such as Korea, China, and Japan, which have tradition of high consumption and production of bivalves, bivalve farms could be located in the proximity of large industrial complexes and cities [7,8,9]. Food safety incidents occurred due to chemical, biological, and physical hazards, and other sources. Chemical hazards are mainly due to pesticides, heavy metals, drug residue, marine toxins, or excessive use of additives. Biological hazards involve contamination by foodborne bacteria, while physical hazards mainly involve non-edible materials, such as plastics, glass, or metal debits [10]. A recent report showed that, in the period between 1998 and 2016, the total number of food safety incidents in Korea was 975, with an average of 51.3 per year [10]. The top two food types involved based on Korean Food Standards Codex 72-2016 were fishery products. Of the 975 incidents, 406 were related to chemical hazards, of which metal contamination accounted for 14.5%. Very relevant for fishery products is that overall primary production was the most common stage where contamination occurred comprising 63.1% of incidents [10].

Metal contamination is a worldwide-recognized public health hazard. Marine organisms, including bivalves, can uptake metals from their environment [11,12] and are, therefore, one of the main routes of exposure of the general population to these elements [13]. It has been estimated that over 90% of human exposure to metal contaminations occurs through diet, primarily seafood and meat [14]. Recently, researchers have focused their attention on metal contents in bivalve species because these animals are widely distributed, and some species can accumulate certain element in higher concentration compared to the others [8]. Chemometric pattern-recognition techniques, such as principal component analysis (PCA) and hierarchical cluster analysis (HCA), have also been applied for evaluation and classification of analytical results, as well as to establish accumulation pattern by organs [15,16].

In this context, the aims of the present study were: (i) to determine the content of essential, non-essential, and toxic elements in the main bivalve species consumed in Korea; (ii) to evaluate species’ ability to accumulate different elements; (iii) to apply PLS-DA as a multivariate classification model method to find mathematical models that can assign each sample to an appropriate species; and (iv) to compare the estimated intakes of each element in each of the species with reference toxicological (risk component) and nutritional values (beneficial component).

## 2. Materials and Methods

### 2.1. Criteria for the Species Selection and Sample Collection

The contribution of the selected species to the total bivalve production in Korea, the location of farms for the given species along the Korean coastline, the importance of the species at the local and global markets, the availability species on Korean market for the consumers during the year, and their frequencies of consumption by consumers were criteria for the selection of species for this study. Among all bivalve species present at the market, according to the criteria, three clams species (*Venerupis philippinarum*, *Anadara broughtonii*, and *Tegillarca granosa*) and two scallops species (*Mizuhopecten yessoensis* and *Argopecten irradians*) were selected.

*Venerupis philippinarum* (Manila clam) is one of the five most commercially valuable bivalve worldwide [17] because of its nutritional value, flavor, and low cost. *Venerupis philippinarum* generally shows high bioaccumulation values for most metals/metalloids, but its role as a bioindicator is still controversial [17]. *Mizohopecten yessoensis* (Yesso scallop) is an economically important bivalve species in the aquaculture and fishery in Asian countries [18]. It is highly valued by consumers and is often consumed fresh. *Argopecten irradians* (the bay scallop) was first introduced from the United States into China in 1982, and it has rapidly become one of the most important marine cultured bivalves in China [19]. Besides *V. philippinarum* and *M. yessoensis*, *Tegillarca granosa* (small blood clam, blood cockle) and *Anadara broughtonii* (big blood clam, blood cockle) are two of eight species, which the most contributed to bivalve production in Korea.

Forty-eight samples (1 to 2 kg per sample) of five selected bivalve species intended for human consumption were purchased from different fishery markets in Korea during April and May 2018. Information on the harvest location of these bivalves was obtained from salesmen.

Taxonomic ranks, common names, and number of samples for each species are summarized in Appendix A.

#### Sample Preparation

After being purchased, live animals were put in the boxes with ice and delivered to the laboratory. The total length and weight of each animal were measured. The whole soft tissue from the shell was removed by the use of a plastic knife. Composite samples were prepared using the tissue samples of minimal 10 animals of the same species. All cleaned and separated tissue samples were cut into pieces and freeze-dried. The dried samples were then crushed using mortar and pestle into a powder and further homogenized before analyses.

Dry samples (1 g) were weighed to a precision of 0.1 mg and placed directly in 12-mL PTFE tubes. Nitric acid (7 mL) and hydrogen peroxide (1 mL) were then added and the samples were digested in the microwave oven (Speedwave, Berghof, Germany). Digestion was performed as follows: samples were heated to 50 °C and held for 10 min at this temperature; next, the temperature was linearly increased to 150 °C and held for 10 min at this temperature; then, the temperature was then linearly increased to 180 °C and held at 180 °C for 20 min. After cooling to room temperature, the digested solutions were transferred to 25-mL volumetric flasks and diluted in distilled water. Each sample was analyzed in duplicate, and each analysis consisted of three replicates.

The mean concentrations of elements (mg/g or µg/g) in the edible parts of the bivalve species were calculated on the wet weight (WW) of samples (Table 1) using water content data for each samples (Appendix A) and determined values of elements in dried samples.

### 2.2. Analysis of Elements

#### 2.2.1. Chemicals, Standards, and Reference Materials

Nitric acid (HNO_3_ 65% p.a., further purified by sub-boiling) and hydrogen peroxide (30% p.a.; were purchased from Merck (Darmstadt, Germany). Water (18.2 MΩ cm) obtained from a Milli-Q water purification system (Millipore Simplicity 185 System with dual UV filters—185 and 254 nm—to remove carbon contamination) was used throughout this work. All glassware was soaked in 4 mol/L HNO_3_ for a minimum of 12 h and rinsed well with distilled water. Multi-element stock solution containing 10.0 mg/L of 22 elements (Alfa Aesar, Haverhill, MA, USA) was used to prepare standard solutions for inductively coupled plasma mass spectrometry (ICP-MS) measurements. The certified reference material DORM-2 (National Research Council of Canada, NRC-CNRC, Ottawa, ON, Canada) was used for the quality control and validation of all measurements.

#### 2.2.2. Instrumentation

Measurements of Cr, Mn, Co, Ni, Cu, Zn, Cd, Pb, As, Hg, and Se were performed using an ICP-Q-MS (iCAP Q X series 2, Thermo Scientific, Winsford, UK) equipped with flat pole collision cell technology (CCT), a micro-concentric nebulizer, platinum cones, and a peristaltic sample delivery pump, running a quantitative analysis mode. High-purity He (99.9999%; Messer, Belgrade, Serbia) was used to minimize potential problems caused by contaminant species in the cell. The collision gas used for Se determination was 7% hydrogen in helium with flow of 3.5 mL/min. The use of hydrogen in the collision cell provided accurate results for Se determination. For total Hg determination, gold solution was added to the samples and the rinse solution to minimize memory effects from the sample introduction system. The entire system was controlled using Qtegra Instrument Control Software (Thermo Scientific, Winsford, UK). Instrumental parameters and measured isotopes are shown in Appendix A. Internal standards ^45^Sc, ^115^In, and ^159^Tb were added to matrix-normalize for all elements.

The measurements of Ca, Mg, K, Na, and Fe were carried out using an Inductively Coupled Optical Emission Spectrometer (ICP-OES; model 6500 Duo, Thermo Scientific, UK) equipped with a CID86 chip detector. This instrument operates sequentially with both radial and axial torch views. The entire system was controlled using the Iteva software. Instrument conditions and selected wavelengths are shown in Appendix A.

Microwave digestion was performed in a Berghof microwave oven (Speedwave, Berghof, Germany). The microwave digestion system was equipped with 12 PTFE vessels.

#### 2.2.3. Quality Control

For quality control and validation of all measurements, the certified reference material DORM-2 was treated and analyzed in the same way as used samples. The results of the analyses were in accordance with the certified levels within a 95% confidence level (Appendix A). Limits of detection (LOD) of Ca, Fe, K, Mg, and Na were 0.45, 0.67, 0.91, 0.19, and 19.31 µg/L in ICP OES, and Cr, Mn, Co, Ni, Cu, Zn, As, Se, Cd, Hg, and Pb were 0.037, 0.042, 0.003, 0.310, 0.058, 0.158, 0.133, 0.136, 0.009, 0.005, and 0.061 µg/L in ICP MS.

### 2.3. Multivariate Analysis

Hierarchical cluster analysis (HCA), and partial least squares discriminant analysis (PLS-DA) were performed using the PLS Toolbox software package (Eigenvector Re- search, Inc., Manson, WA, USA) for MATLAB (Version 7.12.0 R2011a). HCA was performed using the Ward method to calculate Euclidean distance as a measure of distance between samples. PLS-DA was performed using the SIMPLS algorithm without forcing orthogonal conditions to the model in order to condense Y-block variance into first latent variables. PLS-DA is a classical PLS regression in which the response variable is categorical (i.e., it indicates the classes or categories of the samples). Venetian blinds (split ratio 1:5) was used as cross-validation method. The quality of the PLS-DA models was monitored with the several parameters, such as (i) R^2^_cal_, the cumulative sum of squares of the Ys explained by all extracted components, and R^2^_CV_, the cumulative fraction of the total variation of the Ys that can be predicted by all extracted components; these two values should be as high as possible. In addition, root mean square errors of calibration (RMSEC) and root mean square errors of cross-validation (RMSECV) values should be as low as possible, and with the lowest difference between them.

### 2.4. Exposure Assessment and Risk Characterization for Toxic Elements Cd, Pb, and Hg

#### 2.4.1. Preparation of Concentration Data of Elements

For the purpose of an exposure assessment, all provided data recalculated to wet-weight (WW) data were used. Upon evaluation of all data, it became clear that so-called “bound-scenarios” would have to be implemented in the exposure assessment because not all analytical data revealed values > LOD. These scenarios address uncertainty as a result of concentration measurements that fall below the limit of detection (LOD). Three different scenarios are constructed by following these predetermined rules: LB: all measurements below LOD are presumed as being equal to 0; MB: all measurements below LOD are presumed as being equal to LOD/2 and UB: all measurements below LOD are presumed as being equal to LOD. The value of the LOD depends upon the specific element.

#### 2.4.2. Consumption Data

An online food frequency questionnaire survey was organized to collect consumption data of bivalves by South Korean population. Questions on frequency and amount of consumption of different species were elaborated via Survey Monkey^®^. As a validation study, the questionnaire was filled out by 5 persons, to gain information on comprehensiveness of the questions and mode of answering the questions. After this, the online survey was distributed amongst South Korean consumers, by direct e-mailing students and researchers at Ghent University Global Campus, Incheon, Korea, and further spreading via social media amongst the wider Korean population. Therefore, the survey does not represent the South Korean population, as such, but indicates a stratified sample of the South Korean population. The survey was completed by 142 Korean participants in the period February–May 2019 and resulted in the consumption frequency of each particular bivalve species, as well as portion size per meal. Combining this with performed measurements of bivalve sample weights per portion allows for the construction of a consumption distribution for each species.

The survey divides the participants in 9 categories for consumption frequency ranging from never to daily. These results have to be recalculated into daily frequency as to be able to calculate daily exposure to elements. For example: 1 consumption per week = 1/7 consumption per day.

To be able to construct a distribution for consumption frequency, the number of respondents in each category has to be converted into a percentage of total respondents. This information allows the modeling of a discrete distribution in @RISK (Pallisade Corporation, Ithaca, NY, USA) for consumption frequency of the foods. The application of a discrete distribution RiskDiscrete ({x1/x2};{p1/p2}) relies on parameter x, as a frequency of consumption and p, the probability of each respective frequency of consumption occurring.

The survey divides meals of participants into 6 ranges for pieces of bivalve ranging from 0 to >20 to capture the amount of bivalves consumed per meal. The resulting ranges for example “6 to 10 pieces per meal” are applied into @RISK as Pert-distributions which have the following parameters: minimum value, most likely value, and maximum value. For the 6–10 range this implies the function: RiskPert (6;8;10). The Pert-distributions for each range of consumed pieces per meal is multiplied with similarly constructed Pert-distributions for bivalve weight per piece. Combining the distribution for pieces of consumed bivalve per meal with the distribution for weight of those bivalve species results in a new distribution portraying mass of bivalve eaten per meal. To calculate daily consumption (g/person.day), the aforementioned discrete consumption distribution has to be multiplied with the distribution of the weight of bivalve consumed per meal. Exposure to contaminants is expressed on a body weight (BW) basis. To incorporate BW of the Korean population into this exposure assessment, a distribution was constructed using the average South Korean BW from the last 3 Korean National Health and Nutrition Examination Survey (KNHANES) representing years 2016, 2017, and 2018 [20].

#### 2.4.3. Exposure Calculation

A chronic probabilistic exposure assessment is conducted per species and per toxic element Cd, Pb, and Hg. @Risk software (Pallisade Corporation, US, version 8.0), add-in for Excel, was applied to fit distributions to contamination data and for consumption, the distribution as obtained in 2.4.2 was applied. To simulate the exposure, Monte Carlo simulations (100,000 iterations) are used. For the distribution fit to the contamination data, an IF function was introduced in case less than 1/3 of the data points were available above LOD [21,22]. This is the case for Pb in Manila clam, where 10 out of 15 samples where below LOD. Therefore, two functions were introduced: RAND (0,1), so select a random value between 0 and 1 during the iterations and logic IF function (x < y, distribution of contamination, LOD), with y being the fraction of samples <LOD. In this case, LB, MB, and UB scenarios were run to estimate the exposure in case analytical data are <LOD (also see 2.4.1).

#### 2.4.4. Risk Characterization

Based upon literature review, the toxicological reference values for toxic elements Cd, Pb, and Hg were retrieved. A Margin of Exposure (MOE = BMDL (benchmark dose level)/exposure) can be calculated and compared to reference value of 10,000 for carcinogenic compounds [23,24]. In addition, Pb has a BMDL_01_ value of 12 µg/kg BW.day [25], and MOE can be calculated and compared to the reference value of 100 for non-carcinogenic compounds [23]. For Cd, a tolerable weekly intake (TWI) is reported of 2.5 µg/kg BW.week [26]. Finally, Hg has a TWI of 4 µg/kg BW.week [27]. These values are applied in the risk characterization, to compare the exposure values with and to make a conclusion on potential health impact of these toxic elements towards the Korean population. Where possible in this study, BMDL approach was preferred to NOAEL in order to provide better level of protection considering sample size, as well as to provide scientifically more advanced method for deriving a Reference Point, since BMDL makes extended use of available dose-response data, and it provides a quantification of the uncertainties in the dose-response data [24]. Thus, the numerical value of the NOAEL is dependent on the selection of dose levels and on the ability of the study to detect adverse effects. Since studies with low power (e.g., small group sizes) and/or insensitive methods are able to detect only relatively large effects, these tend to result in higher NOAELs [23].

## 3. Results and Discussion

A total of 48 samples of three clam (*Venerupis philippinarum*, *Tegillarca granosa*, and *Anadara broughtonii*) and two scallop (*Mizuhopecten yessoensis* and *Argopecten irradians*) species intended for human consumption, selected in this study, allowed a meaningful biological and statistical interpretation of obtained data. Among all bivalve species present on Korean market, these five species were selected because of their significant contribution into domestic production, and consumption in Korea, as well as their importance for the local and global market. In all samples the content of 4 macro- (Na, K, Mg, and Ca), 8 micro- (Fe, Zn, Mn, Se, Ni, Cu, Cr, and Co), and 4 toxic (As, Cd, Hg, and Pb) elements were determined using ICP MS or ICP OES.

### 3.1. Elements Content in Bivalve Species

#### 3.1.1. Macro and Micro Elements Content

Obtained mean values of macro-elements content (Mg, Na, and K) in the whole-soft tissues were mostly similar in all five species and within the same range as previously published data [28]. The highest variation in mean values of macro element was obtained for Ca. The content of Ca in *T. granosa* was 3.7 times higher than in *M. yessoensis.* These values are similar with results obtained for Ca content in *T.granosa* and *M.yessoensis*, at 71.1 and 22.0 mg/100 g, respectively [28]. The highest variations in mean values of essential elements were obtained for Mn and Ni between two scallop species. The content of Mn in *A. irradians* was 44 times higher than in *M. yessoensis,* while the content of Ni in *M. yessoensis* was 18.5 times higher than in *A. irradians*. Beside these two elements, significant differences between mean values of Co, Zn, and Fe between some species were also found. The highest contents of Fe and Zn were found in *V. philippinarum*, two blood clams (*T.granosa* and *A. broughtonii*), and *A. irradians*, respectively. The average content of Cr, Cu, and Se was similar in all species (Table 1).

The obtained average Mn content (25.1 ± 3.1 µg/g WW) in *A. irradians* was significantly higher than in the other analyzed species. There is lack of available reference data from scientific literature on Mn in whole soft issue in *A.irradians*. The concentrations of Ni in these species were between 0.04 and 0.74 μg/g WW. The average Cu content in the different bivalve species was generally low (0.76–1.39 µg/g WW). Our results within the same range as those of a previous study [14]. Depending on their abilities to eliminate Cu, bivalves contain a lower Cu concentration apart from oysters whose generally contain higher concentrations of Cu [29]. The elimination rates of Cu in scallops and clams are several times high than those of other metals [29]. The Zn content in this study ranged from 8.37 (*V. philippinarum*) to 43 (*A. irradians*) µg/g WW, which is similar to values reported in previous study [1] and does not exceed the maximum value (100 µg/g WW) prescribed by World Health Organization (WHO) [30]. Scallops could accumulate higher level of Zn compare to the other species because of their higher Zn assimilation ability [31].

Conversely, the species from *Pectinidae* family accumulate less Fe than other species. Blood clam carries abundant hemoglobin (Hb) in circulating erythrocytes [32]. As expected, blood clam species, such as *A. broughtonii* and *T. granosa*, have the highest Fe contents (55.8 ± 29.1 and 68.6 ± 23.0 µg/g WW, respectively), which is in accordance with data from the literature [33,34]. Blood clams are listed on prominent healthcare websites as an excellent source of Fe [33]. The concentrations of Cr and Se had a narrow range of variations (0.06–0.10 and 0.35–0.53 μg/g WW, respectively) among species, this is consistent with previous observations for the concentration of Cr in bivalves (0.18–2.97 μg/g) [35].

#### 3.1.2. Toxic Elements Content

Seafood species, including mollusks, contain high levels of As, mostly present as non-toxic organic As compounds, such as arsenobetaine [36]. Therefore, there is a need for speciation data to conduct a risk assessment of As ingested from seafood.

Among all toxic elements Hg was present at the lowest concentration, while the largest variation among investigated species was found for Cd (Table 1). The Hg concentrations in the bivalve tissues were relatively low (under 0.05 μg/g WW, Table 1), and not exceeding the maximum level of 0.5 μg/g established by WHO [37]. It is widely recognized that methyl-Hg (MeHg) can be more easily accumulated in marine organisms than the inorganic forms of Hg [38]. Studies have converted total Hg concentrations to MeHg using a mercury conversion factor (MCF) [39]. The MCFs of 0.5 have been obtained for fish, crustaceans and cephalopods, while MCF of 0.2 was found for bivalves [40]. Interestingly, that molar Se:Hg ratio may be a useful tool to better assess the risk linked with Hg intake [41,42]. Some authors have suggested including this ratio in risk assessment and regulation regarding Hg and fish consumption in humans [43]. High Se content has a protective effect against Hg toxicity as Se may chelate Hg and protect seleno-proteins [44]. According to values obtained from studies on rodents, molar Se:Hg ratio > 1 is adequate to prevent Hg toxicity [45]. The molar Se/Hg ratios were >1 for all samples herein analyzed.

The obtained average Cd contents of investigated species were between 0.11 and 2.05 µg/g WW (Table 1), and great differences between species from different families were found. The highest Cd content was found in *M. yessoensis* (2.05 ± 0.55 µg/g WW). In the European Community, the maximum permitted concentration of cadmium in bivalve molluscs is 1 mg/kg fresh weight [46]. Human consumers generally ingest the whole soft tissues of *M. yessoensis*. However, the total amount of an ingested Cd does not always reflect the amount that is available to the consumer. Therefore, there is necessity to determine the bioaccessibility of toxic metals in shellfish in order to improve health risk assessment. The ability of different *Pectinidae* species to accumulate Cd in soft tissues to a higher degree than other bivalves in nature and under laboratory conditions is well documented [47]. Scallop species, such as *M. yessoensis* and *A. irradians* from *Pectinidae* family, typically contain higher Cd concentrations than other bivalves, which can be attributed to their extremely high Cd assimilation efficiencies and low elimination rates [31]. Probabilistic risk assessment of dietary Cd in the Korean population performed by Kim and Wolt [48] showed that rice, as a commonly and highly consumed single food item by Koreans, is a dominant contributor to Cd in the diet, representing on average 25% of the total dietary exposure for the general population. However, seaweeds, molluscs, and crustaceans tended to have the highest Cd concentrations in Korean foods (averages of 0.3, 0.2, and 0.1 µg/g, respectively), and, cumulatively, seafoods, including shellfish, seaweeds, and fishes, are responsible for about 40% of the total exposure to Cd in the Korean population. In general, exposure assessments through seafood represent a special consideration since, even though average per-capita consumption may be modest (fish, the most consumed seafood item, averages consumption in Korea at 30 g/day), there are always high-end consumers of specific seafood that may dramatically skew exposure estimates.

The Pb concentrations obtained in all five species were very low, from 0.03 to 0.12 μg/g WW, and the below the limit set by WHO [30].

### 3.2. Multivariate Analysisas a Tool to Distinguish Selected Species by Elements Content

To estimate usefulness and efficacy of specific multivariate analyses for distinguishing selected species by elements content, HCA and PLS-DA analysis were performed.

HCA considers all the data variability and shows the similarity/dissimilarity among species based on element concentration [1]. HCA applied to the 48 samples revealed three clusters at 12 variance-weighted distance units: the first and second clusters were entirely comprised of *A. irradians* and *V. philippinarum*, respectively, whereas the third cluster was comprised of samples of the other species (*A.broughtonii*, *M. yessoensis*, *T. granosa*) (Figure 1). The third cluster was probably formed because of the species’ similar metal compositions regarding the high levels of Fe, Cd, Cu, and Pb (Figure 1).

In the present study, PLS-DA, a supervised pattern recognition method, was used to extract maximum information on discriminant features/elements from the data and indicate the elements with highest influence on each species. The model was proposed based on the 48 samples of species and the 16 elements obtained from the metal content analysis. The number of the latent variables (LVs) was selected on the basis of the minimum value of RMSECV, which was achieved with three LVs. Classification and validation results for five models, expressed as R^2^*_cal_*, R^2^*_CV_*, RMSEC, and RMSECV, are presented in Appendix A. The PLS-DA models for *V. philippinarum*, *A. irradians,* and *M. yessoensis* were statistically significant, with relatively high values of R^2^*cal* and R^2^*_CV_*, and low difference between RMSEC and RMSECV values. In contrast, the PLS-DA models for *A. broughtonii* and *T. granosa* had low values of R^2^*cal* and R^2^*_CV_* and low quality classification models (Appendix A). The contributions of elements to the differentiation among species were analyzed using the VIP value. The models show very good results for the classification of *Venerupis philippinarum*, *Argopecten Irradians*, and *Mizuhopecten yessoensis* shellfish samples.

Elements with a VIP score >1 were considered as the most relevant for explaining certain classes of samples model (Figure 2). The VIP scores of the variables show their contribution in the final PLS DA models, with calculated VIP scores above. The certain elements were selected as the most important markers responsible for classification. As, Cd, Co, and Ni were found as taxonomical markers of *V. philippinarum* (Figure 2A); Mn, Zn, Mg, and Na were found as those of *A. irradians* (Figure 2B); and Cd, Ni, and Fe were found as those of *M. yessoensis* (Figure 2C). Significant differences regarding Ni content were found among species, and the highest concentration was found in *V. philippinarum* and *M. yessoensis*. The Cd content was highest in *M. yessoensis*, almost five times the values found in *A. irradians* and *A. broughtonii*. Besides, Mg and Na were found as the most abundant macro-elements in *A.irradians* species and most important for classification.

The standardized regression coefficient reveals the significance of an individual variable in the regression model. Co, As, and Ni contributed most to the PLS-DA model for *V. philippinarum,* while Cd had the most negative contribution to the model for this species (Figure 3A,D). As *V. philippinarum* had the highest and lowest As and Cd contents, respectively, these bioindicators of environmental pollution could be used to distinguish these species from others.

Mn and Zn had the highest positive coefficient in the PLS-DA model for *A.irradians* (Figure 3B,E), which can be explained by these elements being the most accumulated in this species. Further, Cd and Ni had the highest positive contribution on the PLS-DA model for *M. yessoensis* (Figure 3C,F), once again reinforcing the usefulness of this element for distinguishing *M. yessoensis* from the other species. As expected, the PLS-DA model was effective in separating samples according to these three species.

We believe the development and application of this method to a real-world situation will be very effective for detecting fraud, taking into account increasing numbers of mislabeled seafood products [49]. However, for this approach to be successful, the production method (wild or farmed), the geographical origin, and the biological species must be verified and guaranteed. Furthermore, it is necessary to involve a variety of species and different geographic origins.

### 3.3. Deterministic Dietary Intake and Nutritional Point

The meal size or daily ingestion used for the daily intake calculations is different for each country. Portion size refers to the amount of specific food consumed by individual consumers and setting portion size is very important for meal planning, nutrition education, and nutrition assessment. Taking into account the consumers’ habits from different countries, the different portion size can be used during calculations of the potential beneficial effects of bivalve consumption. For example, Bilandžić et al. [50] used the value of 20 g/day as the portion size of Croatia, which is much higher than the portion size of 13.9 g/day used for mollusks in Peru [51]. Based on the data of food intake quantities produced by National Health and Nutrition Examination Survey 2005 for Korean consumers [52], we used a portion of 25-g of clam to calculate element intake.

#### 3.3.1. Essential Elements Cu, Zn, Fe, Mn, and Co

The 25-g portions of the analyzed species contained 0.019 to 0.035 mg of Cu (Appendix A). The Korean Dietary Recommended Intake (RI) set by Korean Nutrition Society (KNS) for adult men and women is 0.80 mg/day [53]. Consumption of a single 25-g portion of the investigated bivalve species provides from 2.4% (*A. irridians and V. philippinarum*) to 4.3% (*M. yessoensis*) of the RDA value. The EFSA Panel on Dietetic Products, Nutrition and Allergies (NDA) derived the following Adequate Intakes (AIs) for Cu: 1.6 mg/day for men and 1.3 mg/day for women (AI for pregnant women = 1.5 mg/day) [54]. The tolerable upper intake level (UL) for adults is 5 mg/day and is not applicable during pregnancy or lactation [55].

The highest content of Zn was found in *A. irradians* (43 µg/g WW; Table 1). The Zn level in all edible bivalve species was below the legal limits (100 µg/g WW) established by WHO [30]. The calculated Zn dietary intake for a 25-g of portion was in range of 0.21–1.06 mg. The Korean Population Reference Intake for Zn for adult males and females is 9.5 mg/day and 8.0 mg/day, respectively [53]; therefore, the daily consumption of a 25-g bivalve portion provides 2.2–11.2% of the RDI reference Zn consumption.

The Korean dietary reference intake (RI) for Fe in all age groups of men and women is 10 mg/day and 12 mg/day, respectively [53]. In the samples of the present study, the highest value (1.71 mg/day) of Fe in 25-g was calculated for the *T. granosa* samples. No UL has been set for Fe by the Scientific Committee for Food (SCF) or EFSA [55].

The 25-g portions of the analyzed species contained in range from 0.014 to 0.63 mg of Mn. *A. irradians* samples contained very high levels of Mn, almost 20-fold higher than in other samples. The Korean dietary adequate intake (AI) for Mn in all age groups of men and women is 3.5 mg/day and 3.0 mg/day, respectively [53]. However, no UL has been set for this element [55].

Finally, Co is an essential trace element as a part of vitamin B12, which is necessary for folate and fatty acid metabolism. There is no clear recommended amount of Co, but there are recommendations for vitamin B12 [56].

#### 3.3.2. Other Elements: Cr, Ni, and Se

The EFSA Panel on Contaminants in the Food Chain (CONTAM Panel) established a tolerable daily intake (TDI) of 300 μg Cr(III)/kg BW per day. However, this evaluation is limited to trivalent Cr because it is the form of Cr that naturally occurs in food [57]. Because of the limited data, a UL was not set. Based on a number of limited studies, it was stated, however, that no evidence of adverse effects associated with supplemental intake of Cr up to a dose of 1 mg/day was found [57]. The portions of bivalve samples herein tested did not contain Cr above this value (less than 2.5 µg per 25-g of portion).

Using the data obtained for Ni in the current study and 25 g of WW of samples as the portion size, we calculated the RDI of Ni. The bivalve portions herein investigated had an amount of Ni lower than the TDI set by EFSA [55] (2.8 μg/kg BW, i.e., 1.2 mg/week for a 60-kg person) (Appendix A).

Moreover, the recommended intake level of 50 µg Se/day was derived for adults [53]. The bivalve portion herein investigated had a lower amount of Se than the RI (found 8.75–13.25 µg/day; or 17.5 to 26.5% of RDI).

### 3.4. Chronic Exposure to Toxic Elements Cd, Pb, and Hg for Korean Population

#### 3.4.1. Consumption Data of *V. philippinarum* and *M. yessoensis* by Korean Consumers

A total of 142 respondents between 20 and 64 years old filled out the food frequency questionnaire on clams and scallop’s consumption. Focusing on these two most important species of this work, Manila clam and Yesso scallop, a distribution expressing the chronic consumption as g/kg BW.day could be made. Appendix A represents the raw data for Yesso scallop out of the survey, applied to construct a discrete distribution on consumption frequency represented in Figure 4.

Further calculation for the number of clams or scallops eaten per consumption moment ranging between 6 and 10 pieces (Pert distribution 6; 8; 10) and the weight per clam or scallop (Pert distribution 37.55; 39.35; 43.04 g) for Yesso scallops with minimum 37.55 g/piece and maximum 43.04 g/piece), resulted in the final chronic consumption distribution. This is illustrated for Yesso scallop in Appendix A, which could finally be corrected with the body weight distribution of Korean population (Pert distribution 60.5; 63.69; 64.3 kg) to express the consumption as g/dayper.kg BW.

#### 3.4.2. Chronic Exposure and Risk Characterization for Korean Population for Cd, Pb and Hg through Consumption of Manila Clam and Yesso Scallop

Based upon the consumption distribution for Manila clam and Yesso scallop and the distribution of contamination to Cd, Pb, and Hg in the foods, the probabilistic exposure was calculated for the Korean population (Table 2), based on 100,000 iterations during the Monte Carlo simulation in @Risk software. For these two species and for the different toxic elements, the calculated probabilistic exposure is ranging from 0 to 1.215 µg/kg BW.per day. The highest value was obtained for Cd in Yesso scallop for the P99 value. It should be noticed that Table 2 represents the UB scenario for Pb in Manila clam, so, where none detects were set as LOD (10 out of 15 analyzed samples), thus representing the worst case scenario of exposure. An illustration of the distribution of the exposure to Cd through consumption of Yesso scallop is given in Appendix A.

Further, different risk characterization outcomes (MOE or TWI) were considered, depending on the element and available toxic reference values. Based on this value, MOE could be calculated for the exposure of P_90_ value (as an example, to cover 90% of the population and not to go into extreme tails of the exposure distribution).

Cd exposure by consumption of Manila clams is not representing a health risk for the Korean population. However, consuming more than a certain amount (248.8 g) of whole scallop tissues per week poses a risk for too high Cd exposure, and 5.3% of the population in the survey had an intake this high. Cd is located in digestive gland that often is removed in large scallop species or could be advised to be removed.

The calculated MOEs for Pb do not indicate a potential health risk, both via consumption of Manila clam and Yesso scallop because calculated values are above 100 (reference MOE for non-carcinogenic compounds). Finally, no exposure above the TWI value for Hg was calculated, indicating the Hg is no health risk via consumption of the two considered species.

In an exposure assessment of elements, typically, concentration data (achieved via analysis of elements in foods) are multiplied with consumption data. However, when doing so, an overestimation of exposure is derived due to the fact that the complete metal content of the food matrix is not available for uptake in the human body. The mitigating factors that constitute this effect are called bioavailability and bioaccessibility. To construct an accurate assessment to any given compound, exposure should be calculated based on the amount of the element likely to be released in the human body, which is the bioaccessible amount, or absorbed, which signifies the bioavailable amount [58]. Reported bioaccessibility ranges for As from 44 to 90.4% [59], Cd from 21.0 to 84%, and Pb from 19 to 86.7% for shellfish [60,61]. It should be emphasized that absorption rates of cadmium are remarkably low compared to cadmium bioaccessibility in seafood products [60]. In case of Pb, recalculation of the exposure distributions and MOE calculation of the exposure at P_90_ value, resulted in an MOE of, respectively, 12,121 for Manila clam and 3141 for Yesso scallop. Comparing these values with the MOEs in Table 2 (MOE, respectively, 5581 and 1448), a clear impact of introducing the bioaccessibility in exposure assessments can be seen.

It can be concluded that there is wide variability between the different evaluated toxic elements. Furthermore, the existence of the substantial differences in bioaccessibility between bivalve species could further increase this variability. Therefore, the reported exposure calculations herein were based on the worst case scenario, which included the UB scenario without including bioaccessibility ranges for the elements in bivalve species.

Moreover, food preparation process allows for reduction of the intake of metals. For example, 29 to 42% As could be removed from rice grains by using an excess of cooking liquid, which is removed after cooking [62]. Other methods of preparation in which moisture loss occurs (e.g., steaming, baking) can also cause a decrease in the As concentration. However, it is important to note that, for some metals, such as As, speciation changes are perhaps more relevant than the total intake [62].

## 4. Conclusions

The strategy of combining metal content in bivalve species with PLS-DA technique allows the identification of factors that can be used in species discrimination. As, Cd, Co, and Ni were found as taxonomical markers of *V. philippinarum*; Mn, Zn, Mg, and Na of *A. irradians*; and Cd, Ni, and Fe of *M. yessoensis*. Revealing taxonomical markers from the elemental content of bivalves could be used for ensuring seafood authenticity regarding geographical origin and species identification. Moreover, the herein presented calculations of possible element dietary intakes showed that these species are good dietary sources of essential elements, especially Mn, Fe, Cu, Zn, and Co. A potential health issue for Korean consumers is related to the intake of the toxic elements present in these bivalve species. The content of two toxic elements Pb and Hg is regulated by the Korean regulatory authority for bivalve species and did not exceed mandated maximum levels in any of the samples. However, the exposure to Cd was found to exceed the toxicological limit for a small percentage (5.3%) of the population through consumption of Yesso scallop. Consuming more than a certain amount (248.8 g) of whole scallop tissues per week poses a risk. Removal of the digestive gland of Yesso scallop before eating will drastically reduce the amount of Cd ingested.

## Figures and Tables

**Figure 1 foods-10-02690-f001:**
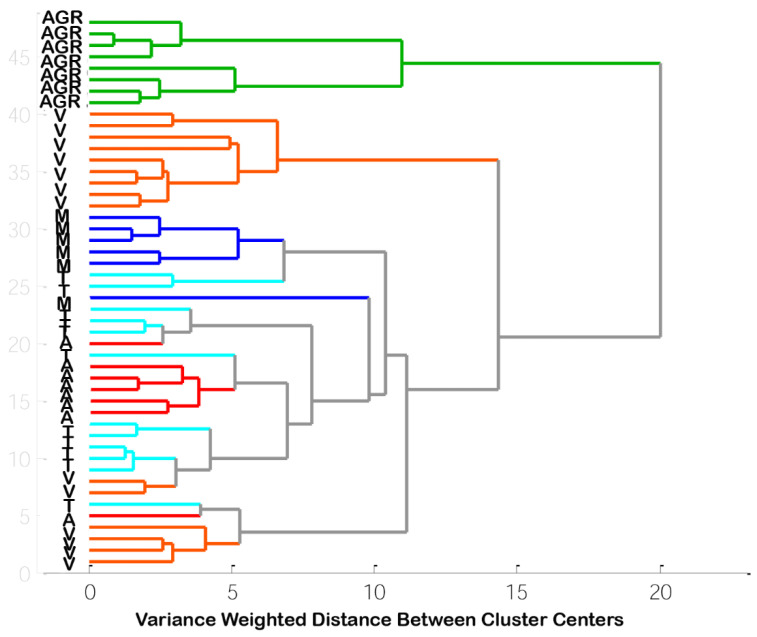
Dendrograms of five taxonomical different bivalve species based on element content (AGR-*Agropectin*
*irradians*, A-*Anadara broughtonii*, M-*Mizuhopecten yessoensis*, T-*Tegillarca granosa*, V-*Venerupis philippinarum*).

**Figure 2 foods-10-02690-f002:**
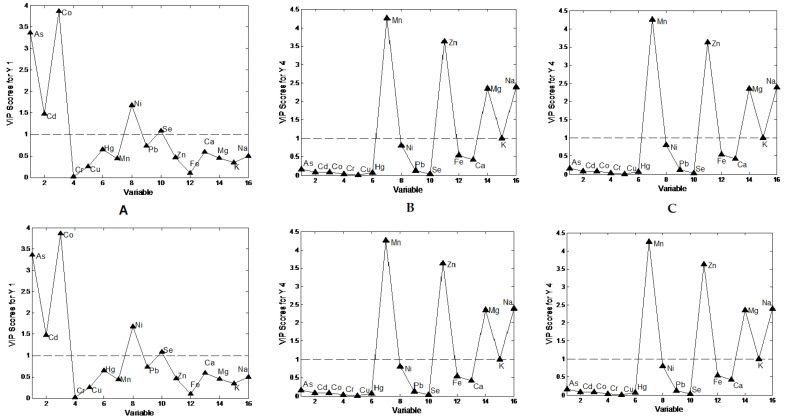
Partial least square-discriminant analysis performed on element analysis: plots of the variables versus VIP scores for: (**A**) *V.philippinarum, (***B**) *A.irradians, (***C**) *M.yessoensis*.

**Figure 3 foods-10-02690-f003:**
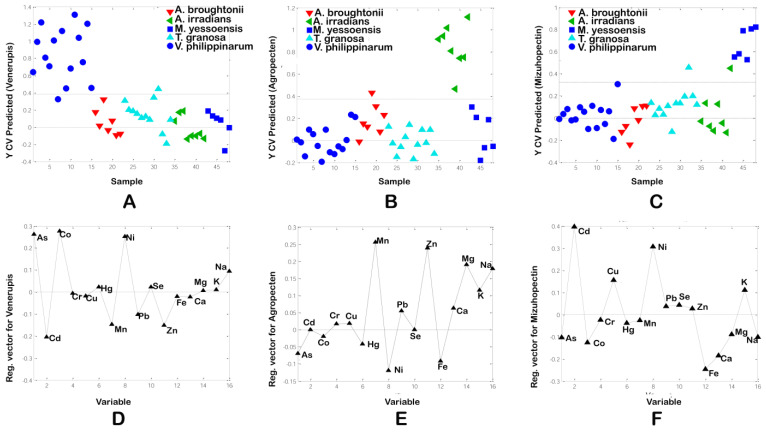
PLS-DA based on element content: scores plots of data and plot of the coefficients of parameters in model for *V.phillipinarum* (**A**,**D**), *A. irradians* (**B**,**E**), and *M. yessoensis (***C**,**F**), respectively.

**Figure 4 foods-10-02690-f004:**
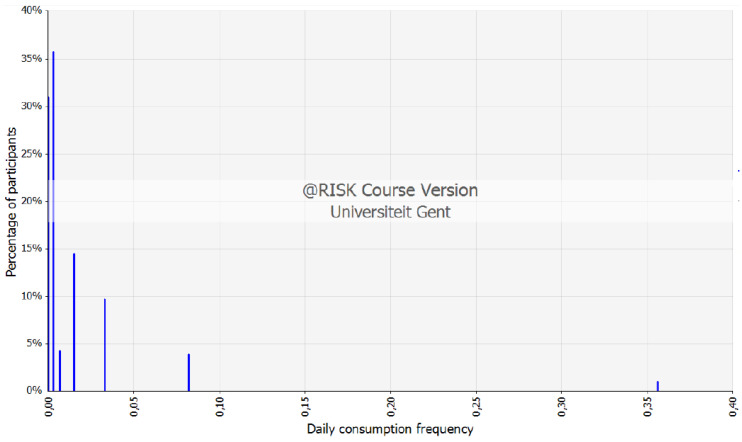
Discrete distribution on frequency of consumption of Yesso scallop (based on the data provided in Appendix A generated via @Risk software).

**Table 1 foods-10-02690-t001:** Content of macro- (mg/g wet weight), micro-, and toxic elements (µg/g wet weight) (mean ± standard deviation) in bivalve species.

Element	Venerupis Philippinarum (*n* = 15)	Anandara Broughtonii (*n* = 7)	Tegillarca Granosa (*n* = 12)	Argopecten Irradians (*n* = 8)	Mizohopecten Yessoensis (*n* = 6)
mg/g					
Ca	0.58 ± 0.22	0.44 ± 0.19	0.75 ± 0.46	0.54 ± 0.15	0.20 ± 0.06
Mg	0.40 ± 0.06	0.56 ± 0.16	0.50 ± 0.10	0.46 ± 0.04	0.39 ± 0.08
K	2.50 ± 0.47	1.98 ± 0.52	2.50 ± 0.60	2.46 ± 0.94	2.50 ± 0.99
Na	3.63 ± 0.49	4.06 ± 0.78	2.95 ± 0.58	4.10 ± 0.56	2.59 ± 1.38
µg/g					
Co	0.15 ± 0.03	0.05 ± 0.03	0.06 ± 0.02	0.07 ± 0.01	0.03 ± 0.01
Cr	0.08 ± 0.03	0.08 ± 0.04	0.10 ± 0.04	0.06 ± 0.04	0.07 ± 0.01
Cu	0.76 ± 0.12	1.06 ± 0.64	1.21 ± 0.59	0.75 ± 0.15	1.39 ± 0.91
Mn	1.32 ± 0.50	3.59 ± 2.32	3.72 ± 1.72	25.15 ± 3.10	0.57 ± 0.14
Ni	0.58 ± 0.18	0.08 ± 0.03	0.12 ± 0.05	0.04 ± 0.01	0.74 ± 0.11
Se	0.52 ± 0.09	0.35 ± 0.14	0.53 ± 0.13	0.37 ± 0.14	0.42 ± 0.03
Zn	8.37 ± 0.98	11.75 ± 3.07	12.46 ± 2.63	43 ± 19	14.5 ± 5.0
Fe	35.0 ± 22.0	55.8 ± 29.1	68.6 ± 23.0	14.0 ± 3.3	15.4 ± 14
Toxic elements					
As	2.61 ± 0.30	1.41 ± 0.77	1.49 ± 0.49	1.07 ± 0.29	0.99 ± 0.23
Cd	0.11 ± 0.04	0.50 ± 0.48	0.86 ± 0.54	0.44 ± 0.21	2.05 ± 0.55
Hg	0.05 ± 0.03	0.03 ± 0.01	0.05 ± 0.03	0.03 ± 0.01	0.03 ± 0.02
Pb	0.03 ± 0.07	0.09 ± 0.07	0.12 ±0.13	0.08 ± 0.04	0.09 ± 0.02

**Table 2 foods-10-02690-t002:** Chronic probabilistic exposure (P_50_ to P_99_ percentile) (expressed as μg/kgBW per day–* UB scenario) to toxic elements (Cd, Pb, and Hg) for the Korean population, by consumption of *V. philippinarum* (Manila clam) and *M. yessoensis* (Yesso scallop) and derived risk characterization, based on available toxicological limits. (MOE = Margin of exposure; TWI = Tolerable weekly intake).

Exposure (μg/kg BW per day)						Risk
Element	Mean ± S.D.	P_50_	P_75_	P_90_	P_99_	Risk Characterization MOE for P_90_ or TWI)
*V. philippinarum* (Manila clam)						
Cd	0.003 ± 0.006	0.001	0.003	0.06	0.024	0% population exceeding TWI = 2.5 μg/kg BW per week
Pb *	0.001 ± 0.004	0	0.001	0.002	0.016	MOE P_90_ with BMDL_01_ 12 µg/kgBW.day = 5581
Hg	0.001 ± 0.003	0.000	0.001	0.003	0.014	0% population exceeding TWI = 4 μg/kg BW per week
*M. yessoensis* (Yesso scallop)						
Cd	0.088 ± 0.402	0.008	0.047	0.178	1.215	5.30% population exceeding TWI = 2.5 μg/kg BW per week
Pb	0.004 ± 0.017	0	0.002	0.008	0.056	MOE P_90_ with BMDL_01_ 12 µg/kgBW.day = 1448
Hg	0.001 ± 0.007	0.000	0.001	0.002	0.017	0% population exceeding TWI = 4 μg/kg BW per week

## Data Availability

The datasets generated for this study are available on request to the corresponding author.

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
