# Peer review of "Chemical Content of Five Molluscan Bivalve Species Collected from South Korea: Multivariate Study and Safety Evaluation"

_foods, 2021, doi:10.3390/foods10112690_

Round 1
Reviewer 1 Report
Authors chose to improve the article by bringing the efforts of the authors and a solution to all criticisms.
Author Response
thank you for comments
Reviewer 2 Report
The manuscript has now been improved enough after major revisions. I have not been able to comment as details as I would like as there are too many comments already.
Introduction
This should be shortened considerably. In particular details regarding production, sales, import etc of bivalves could be omitted in order to keep the focus of the article.
L 69: What kind of incidences are these? Spot checks showing concentrations above regulatory limits? Please elaborate a bit.
Authenticity of seafood products is one of the aims of the study but is not discussed later in the ms. The application of the separation analyses for authenticity or other purposes should be included in the discussion.
Pollution as source of toxic elements in the bivalves is a major part of discussion, and location of farms is one of the criteria of selection of species. Do you have any idea whether the products in the study are from pristine or polluted waters? Are the levels analysed within normal range or elevated due to pollution?
“Chronic probabilistic exposure showed that Cd exposure exceeded the toxicological limit for 5.3% of the population making it a potential health issue”. Regarding scallop? This is not specified
Line 313: Levels similar in all species. But 4 times higher … Mostly similar? Levels: K 100-250 mg/100 g here. 43-190 in ref 35. Na 250-400 vs 120-800. “Within the same range as” could be a better phrasing. Agrees with previously published data for a range of bivalve species…
L 317: Are the values in reference 35 on wet weight and hence comparable with the present study? Hard to tell since the text other than tables is not in English.
L 331: Within the same range..
L 332: Please do not discuss results from literature (like prawns) not relevant to the present study.
L 343: If blood clams in fact contain haemoglobin then this should be included in the discussion with a reference.
L 359: Hg could be deleted here as it is discussed further down. Also Cd
L 360: Check the use of both here.
L 362: The largest variation among the species was found for Cd.
L 396: Only 20% MeHg in bivalves? This may be correct, but this is much lower than in fish.
L 404: Significant differences. How was this tested statistically? Rephrase if not.
L 412 and further: Cd is mostly located in the digestive glands of scallops where extremely high concentrations can be found. Removal of the digestive gland before eating will drastically reduce the amount of Cd ingested and could be an advice in this manuscript. In the EU only adductor muscle and gonads are considered as edible parts (EC 1881/2006). A better phrasing could be that consuming more than a certain amount (calculate this) of whole scallop tissues per week poses a risk for too high Cd exposure, and 5 % of the population in the survey had an intake this high. Removal of the digestive gland will resolve this problem.
L 455: How about the concentrations in the present manuscript? Reference is lacking for the statements here. In this paragraph the concentrations referred to should be discussed in relation to the placement in the PCA, which is lacking in many cases.
L 501: Is the reference 35 relevant to the discussion here with separation of the species by PLS-DA? It seems to refer more to the levels of minerals and not to modelling.
L 571: Consider listing percentages of RDI like above.
Are the samples from clean or polluted sites? Ref intro etc.
I do not have the competence to evaluate the exposure and risk characterization.
L 605: Again, inorganic arsenic is not analysed here or provided from literature and probably low. Discussion of As as toxic element should be deleted. Line 607: Is the BMDL used here for As from the EFSA guidance document referred to in the end of this paragraph? Is this BMDL for organic arsenic or for inorganic arsenic? Delete all discussions regarding toxicity of arsenic unless it is regarding arenobetaine which is dominating in bivalves, fish and most types of seafood of animal origin. This should have been corrected according to my comments to other parts of the document in my previous review.
L 613: Where is the figure 5.3 % calculated and what are the implications? Is this regarding higher intake frequencies or larger portions or a combination? Please elaborate. Same comment as above: Cd is located in digestive gland that often is removed in large scallop species or could be advised to be removed. This should be discussed.
L 629: Discussion of bioaccessibility of inorganic metal forms like Cd and Pb may mislead the reader if absorption is not consideret. As discussed in ref 64: "For the majority of species tested, the absorption of cadmium can range from 0.5% to 3.0% of the dose administered, while in humans a range of 3.0–8.0% can be found. …. These absorption rates are remarkably low compared to cadmium bioaccessibility in seafood products." I would assume that absorption is more important for exposure in the human tissues than bioaccessibility, and absorption rates should be mentioned being much lower.
L 645: Inorganic arsenic is not measured or presented from literature in the present study.
Conclusions:
Review the text here in light of the comments above.
The usefulness of the separation analyses should be addressed, as mentioned above.
L 663: A better phrasing could be that consuming more than a certain amount (calculate this) of whole scallop tissues per week poses a risk for too high Cd exposure, and 5 % of the population in the survey had an intake this high. Removal of the digestive gland will resolve this problem.
Remove text on As unless this is documented for arsenobetaine.
Author Response
Reviewer 2
The manuscript has now been improved enough after major revisions. I have not been able to comment as details as I would like as there are too many comments already.
Introduction
This should be shortened considerably. In particular details regarding production, sales, import etc of bivalves could be omitted in order to keep the focus of the article.
We rewrote introduction and deleted some sentences.
L 69: What kind of incidences are these? Spot checks showing concentrations above regulatory limits? Please elaborate a bit.
We explained what kind of incidents and added sentences: Food safety incidents have occurred due to chemical, biological, and physical hazards, and other sources. Chemical hazards are mainly due to materials such as pesticides, heavy metals, drug residues, marine toxins, or the excessive use of additives. Biological hazards involve contamination by foodborne bacteria while physical hazards mainly involve non-edible materials, such as plastic, glass, or metal debris [10].
Authenticity of seafood products is one of the aims of the study but is not discussed later in the ms. The application of the separation analyses for authenticity or other purposes should be included in the discussion.
We deleted authenticity of seafood from the aims of this study.
Pollution as source of toxic elements in the bivalves is a major part of discussion, and location of farms is one of the criteria of selection of species. Do you have any idea whether the products in the study are from pristine or polluted waters? Are the levels analysed within normal range or elevated due to pollution?
As we pointed out samples were purchased from different fishery markets in Korea intended for human consumption. We obtained information on the harvest location from salesmen but we do not know anything about quality of water.
“Chronic probabilistic exposure showed that Cd exposure exceeded the toxicological limit for 5.3% of the population making it a potential health issue”. Regarding scallop? This is not specified
We corrected into: Cd exposure by consumption of Manila clams is not representing a health risk for the Korean population; however, through consumption of Yesso scallop 5.3% of the Korean population has a potential health risk.
Line 313: Levels similar in all species. But 4 times higher … Mostly similar? Levels: K 100-250 mg/100 g here. 43-190 in ref 35. Na 250-400 vs 120-800. “Within the same range as” could be a better phrasing. Agrees with previously published data for a range of bivalve species…
We corrected it.
L 317: Are the values in reference 35 on wet weight and hence comparable with the present study? Hard to tell since the text other than tables is not in English.
Yes on 100 g wet weight. We carefully check the reference.
L 331: Within the same range..
We corrected it.
L 332: Please do not discuss results from literature (like prawns) not relevant to the present study.
We deleted it.
L 343: If blood clams in fact contain haemoglobin then this should be included in the discussion with a reference.
We added new sentence and new reference: Blood clam carries abundant hemoglobin (Hb) in circulating erythrocytes [32]
Suzuki, T.; Nakamura, A.; Satoh, Y.; Inai, C.; Furukohri, T.; Arita, T. Primary structure of chain I of the heterodimeric hemoglobin from the blood clam Barbatia virescens. J. Protein Chem. 1992, 11, 629–633.
L 359: Hg could be deleted here as it is discussed further down. Also Cd
L 360: Check the use of both here.
We deleted it.
L 362: The largest variation among the species was found for Cd.
We corrected it.
L 396: Only 20% MeHg in bivalves? This may be correct, but this is much lower than in fish.
It is corrected.
You are right. Fish, cephalopods, crustaceans have similar MCF about 50 % (0.5) but shellfish have lower than 20 % (0.2). Greater MCF than 0.5 found only for some predatory fish.
We corrected sentence into: The MCFs of 0.5 have been obtained for fish, crustaceans and cephalopods while MCF of 0.2 was found for bivalves [46]
L 404: Significant differences. How was this tested statistically? Rephrase if not.
We corrected into great differences.
L 412 and further: Cd is mostly located in the digestive glands of scallops where extremely high concentrations can be found. Removal of the digestive gland before eating will drastically reduce the amount of Cd ingested and could be an advice in this manuscript. In the EU only adductor muscle and gonads are considered as edible parts (EC 1881/2006). A better phrasing could be that consuming more than a certain amount (calculate this) of whole scallop tissues per week poses a risk for too high Cd exposure, and 5 % of the population in the survey had an intake this high. Removal of the digestive gland will resolve this problem.
We added paragraph: Consuming more than a certain amount (248.8 g) of whole scallop tissues per week poses a risk for too high Cd exposure, and 5.3 % of the population in the survey had an intake this high. Removal of the digestive gland will resolve this problem.
L 455: How about the concentrations in the present manuscript? Reference is lacking for the statements here. In this paragraph the concentrations referred to should be discussed in relation to the placement in the PCA, which is lacking in many cases.
We deleted PCA paragraph and Figure 1. from manuscript according to suggestion of reviewer 4.
L 501: Is the reference 35 relevant to the discussion here with separation of the species by PLS-DA? It seems to refer more to the levels of minerals and not to modelling.
We deleted this sentence.
L 571: Consider listing percentages of RDI like above.
We added % of RDI
Are the samples from clean or polluted sites? Ref intro etc.
We do not that. We have bought samples from markets.
I do not have the competence to evaluate the exposure and risk characterization.
L 605: Again, inorganic arsenic is not analysed here or provided from literature and probably low. Discussion of As as toxic element should be deleted.
We deleted it as you suggested.
Line 607: Is the BMDL used here for As from the EFSA guidance document referred to in the end of this paragraph? Is this BMDL for organic arsenic or for inorganic arsenic? Delete all discussions regarding toxicity of arsenic unless it is regarding arenobetaine which is dominating in bivalves, fish and most types of seafood of animal origin.
This should have been corrected according to my comments to other parts of the document in my previous review.
We deleted all discussion regarding toxicity of arsenic as you suggested.
L 613: Where is the figure 5.3 % calculated and what are the implications? Is this regarding higher intake frequencies or larger portions or a combination? Please elaborate. Same comment as above: Cd is located in digestive gland that often is removed in large scallop species or could be advised to be removed. This should be discussed.
We added new sentences: Cd is located in digestive gland that often is removed in large scallop species or could be advised to be removed.
L 629: Discussion of bioaccessibility of inorganic metal forms like Cd and Pb may mislead the reader if absorption is not consideret. As discussed in ref 64: "For the majority of species tested, the absorption of cadmium can range from 0.5% to 3.0% of the dose administered, while in humans a range of 3.0–8.0% can be found. …. These absorption rates are remarkably low compared to cadmium bioaccessibility in seafood products." I would assume that absorption is more important for exposure in the human tissues than bioaccessibility, and absorption rates should be mentioned being much lower.
You are right, we added sentence:
It should be emphasized that absorption rates of cadmium are remarkably low compared to cadmium bioaccessibility in seafood products [59].
L 645: Inorganic arsenic is not measured or presented from literature in the present study.
We corrected it.
29 to 42% As could be removed from rice grains by using an excess of cooking liquid, which is removed after cooking [61].
Conclusions:
Review the text here in light of the comments above.
The usefulness of the separation analyses should be addressed, as mentioned above.
We rewrote conclusion part and added this sentence:
Revealing taxonomical markers from the elemental content of bivalves could be used for ensuring seafood authenticity regarding geographical origin and species identification.
L 663: A better phrasing could be that consuming more than a certain amount (calculate this) of whole scallop tissues per week poses a risk for too high Cd exposure, and 5 % of the population in the survey had an intake this high. Removal of the digestive gland will resolve this problem.
We added these sentences: Consuming more than a certain amount (248.8 g) of whole scallop tissues per week poses a risk for too high Cd exposure, and 5.3 % of the population in the survey had an intake this high. Removal of the digestive gland will resolve this problem.
Remove text on As unless this is documented for arsenobetaine.
We deleted all discussion regarding toxicity of arsenic as you suggested.

Reviewer 3 Report
This is straightforward and well-designed study. The information presented in the paper is important for people consuming bivalves. I only few minor suggestions. Title grammar - use "Elemental composition" not "Element composition" (or simply say "Elements as tools for..."). First sentence - can bivalves really be considered as "good source" for carbohydrates?? Methods - add few more details about how Hg and Se analyses were quantified - as these two are more challenging to measure.
Author Response

(The authors gave the same response as above.)

Reviewer 4 Report
The manuscript entitled “Element composition as a tool for the classification of bivalves from South Korean market: health benefits and risks for consumers” describes the determination of a selection of major and trace elements in bivalves used as food in South-Korea and the use of statistical tools for classifying the bivalves in different species based on their element content. Reliable methods in determining trace element in food items are essential and any progress in this direction is interesting. The experimental part is well described and the measurements were carried out properly.
However, there are several important points to improve as listed in detail below. In particular, the concept of the manuscript is unusual as there are two different topics covered: multivariate analysis to find out the bivalves species from the element-pattern, on one side, and the risk assessment of toxic metals for bivalves consumers on the other side. This is giving me the impression that the authors did not want to carry out both aspects completely, and then decided to combine the two incomplete aspects together. Unfortunately, two half built houses are not making one full house. Therefore, I recommend focusing on one of this subject and discuss it till the end. Since the multivariate analysis’ discussion is very short and, to my opinion, not very interesting, I recommend skipping this part or reducing it to a minimum (e.g. cluster analysis to see that species are determining the metal content) and discuss more into details the risk assessment. Alternatively, the authors could also put more efforts in developing the data analysis part. Considering all of this, I recommend major revision.
Please find below more detailed comments:
Title: the title does not make much sense. “Element composition” is not a tool, the classifying tool is the PLS-DA model, for example. Further, the health benefits and risks are not applying to the “element composition” but to the consumption of the bivalves.
Introduction:
I am missing a description of the research gap(s), why is the PLS-DA method innovative? Why is it better than the previously used methods for species identification? For the consumption of bivalves, the levels of metals were already monitored in bivalves in the past; what is the original information delivered in the present study?
L53: “municipal cities” remove municipal
L95-96: the aim (ii) is actually two aims: element content in each specie and use of a multivariate prediction tool.
Material and methods:
L128-129: this sentence can be deleted.
L129: should be “with ice”
L131: correct for “The whole soft tissue was removed from the shell…”
L133: correct for “...were cut into pieces…”
L134: “to make” should be “into”
L142-143: is it correct that the temperature was increased at 190°C and then held at 180°C?
L167: do you mean “in the collision cell”? Otherwise, Ar is used for the plasma generation. But later a mixture of hydrogen gas and helium is mentioned, please clarify.
L182-184: the sentence is redundant with L158-160.
L204-206: what was the size of the validation set?
L217-226: later in the manuscript, only the UB scenario was used. It would be important to mention the proportion of data
L249-252: this is unclear and need more explanation.
L281: define the abbreviations given in this paragraph.
Resuts and discussion:
L299: specify that it is 48 samples per specie.
Table 1: add the number of replicates used to calculate the mean and sd.
L353-358: the first paragraph can be deleted as the information is redundant.
L371: correct the typo
PCA-HCA are data-exploration tools. In this respect, they are not useful to predict the species. Therefore, I recommend to use only HCA to demonstrate the link between specie and element content (PCA is showing the same but less clearly), which is interesting for discussing the exposure later. The predicting tool is the PLS-DA anyway.
Figure 1: the 2D projection renders the 3D-graphs difficult to read. I would recommend using only 2 PCs if the PCA is kept in the manuscript.
An important point would be to look at correlations between the concentrations as some co-occurence of elements may influence the output of multivariate models. Thus, if two variables are highly correlated, they could be merged into one variable hence reducing the number of variables and improving the models.
PLS-DA: since it is the main tool of the paper, I excepted much more quality control and discussion about the performances of the model. For instance, I’m missing a comparison with other mutlivariate tools currently used in the field (decision trees, random forest, neuron networks)) and other methods for determining the specie (e.g. with phenotype or DNA-analysis). Furthermore, the limits of the tool should be discussed. For instance, how can it be calibrated in real control situations? What could be the effect of varying dissolved background in the ocean over the years and locations? Would the integration of further variables such as some morphological traits (e.g. size) improve the prediction for the species for which low R² were obtained?
Paragraph 3.3.1: this part of the discussion is superficial and it is unclear that new knowledge is produced here.
Table 2: this table is not very important and could go in the SI instead of figure S2 which is more important for the discussion. Also figure S1 could be in the main text instead of the PCA results, which are difficult to read anyway.
Table 3: the text is not aligned everywhere.
SI: table S2: was the acquisition time really 50 s ? or rather 50 ms?
There are many typos in the SI, please double check.
Author Response
Reviewer 4
The manuscript entitled “Element composition as a tool for the classification of bivalves from South Korean market: health benefits and risks for consumers” describes the determination of a selection of major and trace elements in bivalves used as food in South-Korea and the use of statistical tools for classifying the bivalves in different species based on their element content. Reliable methods in determining trace element in food items are essential and any progress in this direction is interesting. The experimental part is well described and the measurements were carried out properly.
However, there are several important points to improve as listed in detail below. In particular, the concept of the manuscript is unusual as there are two different topics covered: multivariate analysis to find out the bivalves species from the element-pattern, on one side, and the risk assessment of toxic metals for bivalves consumers on the other side. This is giving me the impression that the authors did not want to carry out both aspects completely, and then decided to combine the two incomplete aspects together. Unfortunately, two half built houses are not making one full house. Therefore, I recommend focusing on one of this subject and discuss it till the end. Since the multivariate analysis’ discussion is very short and, to my opinion, not very interesting, I recommend skipping this part or reducing it to a minimum (e.g. cluster analysis to see that species are determining the metal content) and discuss more into details the risk assessment. Alternatively, the authors could also put more efforts in developing the data analysis part. Considering all of this, I recommend major revision.
Please find below more detailed comments:
Title: the title does not make much sense. “Element composition” is not a tool, the classifying tool is the PLS-DA model, for example. Further, the health benefits and risks are not applying to the “element composition” but to the consumption of the bivalves.
We changed title into: Element contents in five species of bivalves (Mollusca) collected from South Korea: multivariate study and safety evaluation
Introduction:
I am missing a description of the research gap(s), why is the PLS-DA method innovative? Why is it better than the previously used methods for species identification? For the consumption of bivalves, the levels of metals were already monitored in bivalves in the past; what is the original information delivered in the present study?
We added sentences into introduction: Chemometric pattern-recognition techniques such as principal component analysis (PCA) and hierarchical cluster analysis (HCA) have also been applied for evaluation and classification of analytical results as well as to establish accumulation pattern by organs [15,16].
Silva,E.; Viana, Z. C. V.; Souza, N. F. A.; Korn, M. G. A.;V. Santos, L. C. S. Assessment of essential elements and chemical contaminants in thirteen fish species from the Bay Aratu, Bahia, Brasil, Braz. J. Biol., 2016, 76, 871-877. http://dx.doi.org/10.1590/1519-6984.02415
Silva,E.; Costa,F.N.; Souza, T. L.; Viana, Z. C. V; Souza, A. S.; Korn, M.G. A. Ferreira, S. L. C. Assessment of Trace Elements in Tissues of Fish Species: Multivariate Study and Safety Evaluation, J. Braz. Chem. Soc., 2016, 27, 2234-2245. http://dx.doi.org/10.5935/0103-5053.20160116
L53: “municipal cities” remove municipal corrected.
L95-96: the aim (ii) is actually two aims: element content in each specie and use of a multivariate prediction tool.
We rephrased aims into:
In this context, the aims of the present study were: (i) to determine the content of essential, non-essential, and toxic elements in the main bivalve species consumed in Korea; (ii) to evaluate species’ ability to accumulate different elements; (iii) to apply PLS-DA as a multivariate classification model method to find mathematical models that can assign each sample to an appropriate species and (iv) to compare the estimated intakes of each element in each of the species with reference toxicological (risk component) and nutritional values (beneficial component).
Material and methods:
L128-129: this sentence can be deleted. corrected.
L129: should be “with ice” corrected.
L131: correct for “The whole soft tissue was removed from the shell…” corrected.
L133: correct for “...were cut into pieces…” corrected.
L134: “to make” should be “into” corrected.
L142-143: is it correct that the temperature was increased at 190°C and then held at 180°C? corrected.
L167: do you mean “in the collision cell”? Otherwise, Ar is used for the plasma generation. But later a mixture of hydrogen gas and helium is mentioned, please clarify.
We added sentences: This instrument uses a Kinetic Energy Discrimination (KED) mode to eliminate polyatomic interferences by collisions with helium gas. The collision gas used for Se determination was 7% hydrogen in helium with flow of 3.5 mL/min. The use of hydrogen in the collision cell provided accurate results for Se determination.
L182-184: the sentence is redundant with L158-160. corrected
L204-206: what was the size of the validation set?
In current study, we did not split set into training and test sets. This is qualitative approach.
L217-226: later in the manuscript, only the UB scenario was used. It would be important to mention the proportion of data
We checked the data again, and the UB scenario is only applied for Pb in Manila clam. All the other elements (Cd, Hg, Pb) and the other species (Yesso scallop) samples where above LOD and did not need this treatment. Here, no IF function was needed and a direct fitting of the data was possible to establish a distribution. This has been clarified in lines 244 and table 2 and lines 500-501.
L249-252: this is unclear and need more explanation.
The application of a discrete distribution RiskDiscrete ({x1/x2};{p1/p2}) relies on parameter x, as a frequency of consumption and p, the probability of each respective frequency of consumption occurring.
L281: define the abbreviations given in this paragraph.
We added benchmark dose level, BMDL, It is given also in the list of abbreviations
Resuts and discussion:
L299: specify that it is 48 samples per specie. corrected
Table 1: add the number of replicates used to calculate the mean and sd.
Each sample was analyzed in duplicate, and each analysis consisted of three replicates.
In table 1 are given mean values and standard deviation between samples. We added number of samples per species. They are also presented in Table S1 (Supplementary material)
L353-358: the first paragraph can be deleted as the information is redundant.
We deleted the first paragraph.
L371: correct the typo corrected
PCA-HCA are data-exploration tools. In this respect, they are not useful to predict the species. Therefore, I recommend to use only HCA to demonstrate the link between specie and element content (PCA is showing the same but less clearly), which is interesting for discussing the exposure later. The predicting tool is the PLS-DA anyway.
Figure 1: the 2D projection renders the 3D-graphs difficult to read. I would recommend using only 2 PCs if the PCA is kept in the manuscript.
We deleted PCA paragraph and Figure 1. from manuscript according to your suggestion
An important point would be to look at correlations between the concentrations as some co-occurence of elements may influence the output of multivariate models. Thus, if two variables are highly correlated, they could be merged into one variable hence reducing the number of variables and improving the models.
PLS-DA: since it is the main tool of the paper, I excepted much more quality control and discussion about the performances of the model. For instance, I’m missing a comparison with other mutlivariate tools currently used in the field (decision trees, random forest, neuron networks)) and other methods for determining the specie (e.g. with phenotype or DNA-analysis). Furthermore, the limits of the tool should be discussed. For instance, how can it be calibrated in real control situations? What could be the effect of varying dissolved background in the ocean over the years and locations? Would the integration of further variables such as some morphological traits (e.g. size) improve the prediction for the species for which low R² were obtained?
Thank you for your comments. We added several sentences regarding to PLS DA. Due to small size of data set, (Anandara broughtonii (n=7), Tegillarca granosa (n=12), Argopecten irradians (n=8), Mizohopecten yessoensis (n=6)) we did not apply decision trees, random forest, neuron networks. This research is preliminary study with qualitative approach od PLS DA and limited number of samples, because of that we did not split data into training and test set.
Paragraph 3.3.1: this part of the discussion is superficial and it is unclear that new knowledge is produced here.
In our opinion it should be keep this paragraph because of aims of this study not only risk but benefits as well.
Table 2: this table is not very important and could go in the SI instead of figure S2 which is more important for the discussion. Also figure S1 could be in the main text instead of the PCA results, which are difficult to read anyway.
We transferred Table 2 to Supplementary material as you suggested and add Fig S2 in the main text as Fig. 3.
Table 3: the text is not aligned everywhere.
We corrected it.
SI: table S2: was the acquisition time really 50 s ? or rather 50 ms?
There are many typos in the SI, please double check.
We corrected all typos in SI.

Round 2
Reviewer 4 Report
The authors carried out significant improvements on the manuscript. I only have some remarks:
PLS-DA model: is the data were not split into training and test sets, how could the cross-validation errors be determined? (see lines 183-184)
Even if it is described as “qualitative”, PLS-DA is by essence a predictive model and, therefore, it is important to check if the prediction works properly. Otherwise, the evaluation of the input variables’ importance would be of poor quality as, for instance, overfitting cannot be detected without cross-validation. Maybe the authors could find a work around by splitting the small data set into training and test sets (can be low like 90-10%) and carry out the model on many different random partitions (e.g. 100). This way, the stability of the model could be evaluated despite the low number of observations. This would be just to check the quality of the model, the final classification tool could be build using all data.
In addition, I think the discussion would benefit from a more detailed discussion about the application of the method to real situations e.g. for detecting frauds. What would be the advantages and limits of the method and what are the next steps required for developing further this approach?
Line 398: should be “for classification”.
Author Response
The authors carried out significant improvements on the manuscript. I only have some remarks:
PLS-DA model: is the data were not split into training and test sets, how could the cross-validation errors be determined? (see lines 183-184)
Even if it is described as “qualitative”, PLS-DA is by essence a predictive model and, therefore, it is important to check if the prediction works properly. Otherwise, the evaluation of the input variables’ importance would be of poor quality as, for instance, overfitting cannot be detected without cross-validation. Maybe the authors could find a work around by splitting the small data set into training and test sets (can be low like 90-10%) and carry out the model on many different random partitions (e.g. 100). This way, the stability of the model could be evaluated despite the low number of observations. This would be just to check the quality of the model, the final classification tool could be build using all data.
Thank you very much for your suggestion. We are afraid that you might misunderstood us. We have already mentioned in the manuscript, that we used a cross-validation (Lines 179-184). Now, we have added one more sentence regarding the cross-validation strategy (Lines 178-179). The aim of the paper was not development of a predictive model, but rather descriptive one (explanatory), that is of qualitative nature and because of that we did not split data into separate external (test) and training set. Venetian blinds resampling strategy with object split ratio 1:5, was used as cross validation method to assess the overall statistical quality of our models and find optimal model complexity. Also, in the Supplementary in the Table S5 we summarized all statistical parameters of models, including those based on CV such as R2CV and root mean square errors of cross-validation (RMSECV).
In addition, I think the discussion would benefit from a more detailed discussion about the application of the method to real situations e.g. for detecting frauds. What would be the advantages and limits of the method and what are the next steps required for developing further this approach?
We added new sentences (Lines 421-426) and new reference.
We believe the development and application of this method to a real-world situation will be very effective for detecting fraud, taking into account increasing numbers of mislabeled seafood products [49]. However, for this approach to be successful, the production method (wild or farmed), the geographical origin, and the biological species must be verified and guaranteed. Furthermore, it is necessary to involve a variety of species and different geographic origins.
Ortea, I.; Gallardo, J.M. Investigation of production method, geographical origin and species authentication in commercially relevant shrimps using stable isotope ratio and/or multi-element analyses combined with chemometrics: an exploratory analysis. Food Chem. 2015, 170, 145‐153. https://doi.org/10.1016/j.foodchem.2014.08.049.
Line 398: should be “for classification”. Corrected

This manuscript is a resubmission of an earlier submission. The following is a list of the peer review reports and author responses from that submission.
Round 1
Reviewer 1 Report
The manuscript has an interesting approach and parts of it is well written. However, there are some serious concerns, in particular regarding the use of references without relevant content, and hence I have only performed a partly review.
Materials and methods: Common names or type of bivalve for the two last species is missing. L 313 states blood clams.
L 203: More simple: Recalculated to ww. Repeated in results line 301, use this short form in materials and methods.
Why are only clams and scallops selected? Oysters and mussels are not important in Korea?
“Obtained mean values of macro-elements content were similar in all five species and they are in agreement with previously published values [35]”. Does the reference list values in bivalves? Could not find this.
L 316: “The obtained average Mn content (25.1± 3.1 µg/g WW) in A. irradians was signifi-
317 cantly higher than in the other species, and this is in agreement with results of previously
318 published study [12]”. The reference does not include these species.
“The average Cu content in the different bivalve species was generally low (0.76–1.39 µg/g WW). Our results are in agreement with those of a previous study [15]. The highest concentrations of Cu were found in shrimp and prawn species as they have hemocyanin, a copper-containing protein that functions as an oxygen-transport molecule.” This reference only includes Ruditapes philippinarum compared to shrimps and other types of seafood. I cannot see that this reference goes into details of concentrations of Cu in clams vs shrimps. I would anyway question the relevance of this comparison.
Are the scallops analyses whole without removal of the hepatopancreas? In particular Mizohopecten yessoensis can be large and the whole soft tissues will probably not be eaten, in particular the digestive gland with extremely high Cd concentrations has a bad taste. The high Cd values indicate that this is not edible parts as adductor muscle and gonad.
L 341…Total As should not be evaluated since it is the inorganic arsenic that has the cancerogenic effect. This part should be omitted.
L 353: This does not make sense: “The mean concentrations of As were from 0.99–2.61 µg/g WW, and in all samples were below the limit set by WHO (inorganic As 1.0 mg/kg WW) [41]”. As discussed in the ms, values of total As cannot be evaluated as risk as the toxic form iAs cannot be predicted. Cut the discussion of As.
The Hg concentrations in the bivalve tissues were relatively low (under 0.05 μg/gWW, Table 1), and not exceeding the maximum level of 0.5 μg/g established by WHO (2011) [34]. The reference does not mention merury.
The same is the case for the reference for Cd but now with refence to number 54, which is not by WHO. The WHO reference does not give a maximum level but refers to the EU MLs for cadmium and others in Commission Regulation (EC) No 1881/2006, which for Cd in bivalves is 1.o mg/kg, and not 2.0 µg/g as stated in the ms (the safety level of 2.0 µg/g WW established by WHO (2011) [54]).
“Reported bioaccessibility ranges for As from 44 to 90.4 % [65], Cd from 21.0 to 84%, Pb from 19 to 86.7% for shellfish [66,67]”. Ref 66 is “Scientific opinion on dietary reference values for chromium”. Ref 67 fulltext is only available in Dutch and French. The availability of an inorganic element like Cd should be much lower than 21-84 %, so there is something fundamentally wrong here.
I cannot check all references and conclude that there are serious problems with the way references are used in this manuscript.
Line 593: “Cd exposure by consumption of Manila clams is not representing a health risk for the Korean population; however, through consumption of Yesso scallop 5.3% of the Korean population has a potential health risk”. I do not have the competence for reviewing the risk characterization. However, as mentioned above, the analysis of the Yesso scallop is probably done on whole soft tissue including the high-Cd digestive gland and does probably not represent the edible parts. The risk characterization of Cd in the Yesso scallop would hence not be valid and would over-estimate the Cd-intake from these scallops.
Author Response
Manuscript
Title: Element composition as a tool for the classification of bivalves from South Korean market: health benefits and risks for consumers
Referee’s response with comments and action of the authors
Our action, responses and comments are given in bold
Reviewer 1:
The manuscript has an interesting approach and parts of it is well written. However, there are some serious concerns, in particular regarding the use of references without relevant content, and hence I have only performed a partly review.
Materials and methods: Common names or type of bivalve for the two last species is missing. L 313 states blood clams.
We added common names:
Beside V. philippinarum and M. yessoensis, Tegillarca granosa (small blood clam, blood cockle) and Anadara broughtonii (big blood clam, blood cockle) are two of eight species, which the most contributed to bivalve production in Korea.
L 203: More simple: Recalculated to ww. Repeated in results line 301, use this short form in materials and methods.
We corrected it.
Why are only clams and scallops selected? Oysters and mussels are not important in Korea?
There are several reasons why oysters and mussels were not selected in this study. Oyster (Crassostrea gigas) makes the largest contribution to Korean aquaculture production, but it is not on the first place according to the consumption (Table below). Reason for this discrepancy is that 90 % of oyster`s production is exported to other countries. Manila clam is on the second place according to the contribution to the aquaculture production and it is on the first place according to the consumption by Korean consumers. The oyster season in South Korea lasts from the end of October to May, so during the collection of samples for this study sufficient number of samples of oysters were not available on the market (Please, see the criteria for species selection, Materials and Methods 2.1.). Among mussel`s species, native species like Mytilus edulis and Mytilus coruscus are more valuable for the consumers. The invasive species, Mytilus galloprovincialis was introduced into Korea several decades ago and this species have been conquered the habitats of domestic mussels species during this period. Because of this fact, currently mussels are commonly sold as a mixture of these species on the Korean market.
Species |
Common name |
MT |
% |
Crassostrea gigas |
Pacific cupped oyster |
239270 |
78,5 |
Ruditapes philippinarum |
Manila clam Little neck clam |
27570 |
9 |
Mytilus galloprovancialis |
Mussel |
20409 |
6.7 |
Tegillarca (Anadara) granosa |
Small Blood Arc |
10849 |
3.6 |
Anadara broughtonii |
Giant Blood Arc |
3134 |
1 |
Atrina pectinata |
Pen Shell |
1997 |
0,7 |
Haliotis discus discus |
Abalone (snail) |
1260 |
0.4 |
Patinopectin yessoensis |
Yesso Scallop |
173 |
0,1 |
Maretrix lusoria |
Hard clam |
127 |
<0.1 |
Total |
|
304 889 |
100 |
“Obtained mean values of macro-elements content were similar in all five species and they are in agreement with previously published values [35]”. Does the reference list values in bivalves? Could not find this.
We would like to apologize for the mistakes with the references. It was a technical issue, this was due to the format change between our manuscript word and the article format. The doi number on the first reference created a new line, this resulted in a shift of numbering. Therefore, all the references are number incorrectly.
L 316: “The obtained average Mn content (25.1± 3.1 µg/g WW) in A. irradians was signifi-
317 cantly higher than in the other species, and this is in agreement with results of previously
318 published study [12]”. The reference does not include these species.
We corrected it.
“The average Cu content in the different bivalve species was generally low (0.76–1.39 µg/g WW). Our results are in agreement with those of a previous study [15]. The highest concentrations of Cu were found in shrimp and prawn species as they have hemocyanin, a copper-containing protein that functions as an oxygen-transport molecule.” This reference only includes Ruditapes philippinarum compared to shrimps and other types of seafood. I cannot see that this reference goes into details of concentrations of Cu in clams vs shrimps. I would anyway question the relevance of this comparison.
We corrected reference number in the list of references.
Are the scallops analyses whole without removal of the hepatopancreas? In particular Mizohopecten yessoensis can be large and the whole soft tissues will probably not be eaten, in particular the digestive gland with extremely high Cd concentrations has a bad taste. The high Cd values indicate that this is not edible parts as adductor muscle and gonad.
In Korea, Japan and China Mizohopecten yessoensis is very valuable bivalve. It can be eaten raw as sushi or a snack while drinking alcohol or it may be cooked for use in various dishes. When it is eaten raw the digestive system track the hepatopancreas is removed, while other parts of scallops like adductor muscle, gland and the mantle are used. When it is used cooked then the whole animal is eaten. In our study, we used whole scallops for analysis as the worst exposure scenario.
L 341…Total As should not be evaluated since it is the inorganic arsenic that has the cancerogenic effect. This part should be omitted.
We deleted these sentences.
L 353: This does not make sense: “The mean concentrations of As were from 0.99–2.61 µg/g WW, and in all samples were below the limit set by WHO (inorganic As 1.0 mg/kg WW) [41]”. As discussed in the ms, values of total As cannot be evaluated as risk as the toxic form iAs cannot be predicted. Cut the discussion of As.
According to your suggestion we cut the discussion of As and left only two sentences
“Seafood species, including mollusks, contain high levels of As but mostly present as non-toxic organicAs compounds such as arsenobetaine [43]. Therefore, there is a need for speciation data to conduct a risk assessment of As ingested from seafood.”
In our opinion, it is important to emphasize necessity of speciation analysis.
The Hg concentrations in the bivalve tissues were relatively low (under 0.05 μg/gWW, Table 1), and not exceeding the maximum level of 0.5 μg/g established by WHO (2011) [34]. The reference does not mention mercury.
We corrected reference number in the list of references.
The same is the case for the reference for Cd but now with reference to number 54, which is not by WHO. The WHO reference does not give a maximum level but refers to the EU MLs for cadmium and others in Commission Regulation (EC) No 1881/2006, which for Cd in bivalves is 1.o mg/kg, and not 2.0 µg/g as stated in the ms (the safety level of 2.0 µg/g WW established by WHO (2011) [54]).
We corrected sentence and added new reference in the list of references.
“Reported bioaccessibility ranges for As from 44 to 90.4 % [65], Cd from 21.0 to 84%, Pb from 19 to 86.7% for shellfish [66,67]”. Ref 66 is “Scientific opinion on dietary reference values for chromium”. Ref 67 fulltext is only available in Dutch and French.
We corrected reference number in the list of references.
The availability of an inorganic element like Cd should be much lower than 21-84 %, so there is something fundamentally wrong here.
As we wrote, these are reported values. These values are Bioaccessibility not availability.
Bioavailability and bioaccessibility are often misunderstood concepts; although related, they have different definitions. Whereas bioaccessibility is the amount of a determined compound that is released from a matrix becoming available to be absorbed after undergoing the digestion process, bioavailability consists of the amount of a compound that reaches the systemic circulation and exerts its effect after being metabolized and distributed by tissues (https://doi.org/10.1021/acsomega.8b03499). We agree that cadmium absorption after dietary exposure in humans is relatively low (3–5 %).
I cannot check all references and conclude that there are serious problems with the way references are used in this manuscript.
We corrected reference number in the list of references.
Line 593: “Cd exposure by consumption of Manila clams is not representing a health risk for the Korean population; however, through consumption of Yesso scallop 5.3% of the Korean population has a potential health risk”. I do not have the competence for reviewing the risk characterization. However, as mentioned above, the analysis of the Yesso scallop is probably done on whole soft tissue including the high-Cd digestive gland and does probably not represent the edible parts. The risk characterization of Cd in the Yesso scallop would hence not be valid and would over-estimate the Cd-intake from these scallops.
As we explained Mizohopecten yessoensis can be eaten raw as sushi/a snack or it may be cooked for use in various dishes. When it is eaten raw the digestive system track is removed, but when it is used cooked then the whole animal is eaten. In our study, we used whole scallops for analysis as the worst exposure scenario.
We are grateful to you and all referees in your efforts to improve our manuscript.

Reviewer 2 Report
The Abstract should do a better job at summarizing the main point of the paper. Currently it provides too much detail, while at the same time not enough information about the results of the study. Main results and conclusion should be given here.
There is no Rf. D. value in the new recommendations for lead and compounds (see Regional Screening Level (RSL) Summary Tables in 2021 and the Risk Assessment Information System in 2021). In contrast, oral slope factor (SF) is given for lead and compounds as 0.0085 mg/kg/day. Risk analysis (RI) is also calculated with the formula RI = EDI x SF.
The maximum allowable values of elements such as Pb, Hg and Cd, which have carcinogenic effects, are given as 0.3, 0.5 and 0.05 mg / kg wet wt., respectively, in the European Union Commission Regulation. In this study, Cd was found (0.11 and 2.05) to be well above the acceptable values. Please explain / discuss this to readers.
The literature in references [54] is inconsistent with the WHO (2011) given on Line 394.
Line 761 it was written "32. 30. European Food Safety Authority. Update: use of the benchmark dose approach in risk assessment. EFSA J. 2017, 15(1), 465" This is 32 or 30th literature?
The literature, both in the whole text and in the references, should be carefully reviewed according to the journal rules and written correctly.
Author Response
Manuscript
Title: Element composition as a tool for the classification of bivalves from South Korean market: health benefits and risks for consumers
Referee’s response with comments and action of the authors
Our action, responses and comments are given in bold
Reviewer 2
The Abstract should do a better job at summarizing the main point of the paper. Currently it provides too much detail, while at the same time not enough information about the results of the study. Main results and conclusion should be given here.
We rephrased and wrote new abstract.
There is no Rf. D. value in the new recommendations for lead and compounds (see Regional Screening Level (RSL) Summary Tables in 2021 and the Risk Assessment Information System in 2021). In contrast, oral slope factor (SF) is given for lead and compounds as 0.0085 mg/kg/day. Risk analysis (RI) is also calculated with the formula RI = EDI x SF.
We did a probabilistic risk assessment, not deterministic and applying: RI : EDI x SF. Risk assessment calculations could be calculated in many ways but we opted for the full exposure assessment calculations as recommended by EFSA, to exploit the available data fully and include variability.
The maximum allowable values of elements such as Pb, Hg and Cd, which have carcinogenic effects, are given as 0.3, 0.5 and 0.05 mg / kg wet wt., respectively, in the European Union Commission Regulation. In this study, Cd was found (0.11 and 2.05) to be well above the acceptable values. Please explain / discuss this to readers.
We added new paragraph:
In the European Community, the maximum permitted concentration of cadmium in bivalve molluscs is 1 mg/kg fresh weight (COMMISSION REGULATION (EC) No 1881/2006). Human consumers generally ingest the whole soft tissues of M. yessoensis. However, the total amount of an ingested Cd does not always reflect the amount that is available to the consumer. Therefore, there is necessity to determine the bioaccessibility of toxic metals in shellfish in order to improve health risk assessment.
The literature in references [54] is inconsistent with the WHO (2011) given on Line 394.
We would like to apologize for the mistakes with the references. It was a technical issue; this was due to the format change between our manuscript word and the article format. The doi number on the first reference created a new line; this resulted in a shift of numbering. Therefore, all the references are number incorrectly.
Line 761 it was written "32. 30. European Food Safety Authority. Update: use of the benchmark dose approach in risk assessment. EFSA J. 2017, 15(1), 465" This is 32 or 30th literature?
We corrected it.
The literature, both in the whole text and in the references, should be carefully reviewed according to the journal rules and written correctly.
We completely corrected all errors.
We are grateful to you and all referees in your efforts to improve our manuscript.

Round 2
Reviewer 1 Report
I was happy to learn that the numbering in the list of references could be the cause of the poorly suitable references commented in the previous review. However, after checking two of the references in the revised manuscript I still do not find the use of references here satisfactory:
“Obtained mean values of macro-elements content were similar in all five species and they are in agreement with previously published values [35]”. Five bivalves from Brazil. And no values in mg/kg. Not suitable for comparison.
L 316: “The obtained average Mn content (25.1± 3.1 µg/g WW) in A. irradians was signifi-
317 cantly higher than in the other species, and this is in agreement with results of previously
318 published study [12]”. The reference includes A purpuratus and does not contain analyses of Mn or other minerals in whole soft tissue. How did you do the comparison with your data? You also seem to confirm the finding that Mn was higher in the scallop compared to the other species of bivalves. A publication of levels in a scallop but no other species of bivalves is not sufficient for this comparison.
I will not proceed with commenting the manuscript. Pleas check thoroughly the use of references before sending a revised version of the manuscript.
Reviewer 2 Report
Authors made the article more understandable by making the deficiencies and the specified corrections. Now the article has become more advanced.